# Drug Use among the Elderly Assisted by the Psychosocial Assistance Center in District Federal-Brasilia

**DOI:** 10.3390/healthcare10060989

**Published:** 2022-05-26

**Authors:** João de Sousa Pinheiro Barbosa, Leonardo Costa Pereira, Marileusa Dosolina Chiarello, Kerolyn Ramos Garcia, Giovanna Oliveira de Brito, Eliana Fortes Gris, Margô Gomes de Oliveira Karnikowski

**Affiliations:** 1Faculty of Education and Health Sciences—FACES—Medicine Course, University Center of Brasília—UniCEUB, 707/907, Asa Norte, Brasilia 70790-075, Brazil; jspb06@gmail.com; 2Euroamerican University Center—UniEURO, Avenida das Nações, Trecho 0, Conjunto 05, Asa Sul, Brasilia 70200-001, Brazil; 3Department of Pharmacy, University of Brasília, Brasilia 72220-275, Brazil; marileusa.chiarello@gmail.com; 4Graduate Program in Health Sciences and Technologies, Faculty of Ceilandia, University of Brasilia, Centro Metropolitano, Conjunto A, Lote 01, Brasilia 72220-275, Brazil; kerolynramos@gmail.com (K.R.G.); giovanna_.b1@hotmail.com (G.O.d.B.); elianagris@gmail.com (E.F.G.); margounb@gmail.com (M.G.d.O.K.)

**Keywords:** aged, substance related disorders, substance misuse, frail elderly community-based treatment, Brazil

## Abstract

The inadequate and abusive usage of psychoactive substances is something real that affects Brazil’s elderly population, and it is a huge challenge for the public health system and its professionals. Aware of the social impact involving the use of illegal drugs, in 2002, the Ministry of Health instituted a network of psychosocial assistance as a strategy to deal with the problem. This study carried out an analysis of the profile of use of legal and illegal drugs by the elderly who are assisted by the network of psychosocial assistance in the Federal District. A quantitative and analytical study with secondary data collection, using patient records held in the CAPS-AD in the Federal District. The inclusion criteria were people of 60 and over who were users of alcohol and other drugs and who sought assistance at CAPS-AD between 2000 and 2017. A total of 408 medical records were analyzed concerning social demographic variations, types of rehabilitation services sought, types of substances consumed, associations between drugs consumed, time of consumption, and adherence to the treatment. Most of the elderly users were male (85.3%), on average 64 ± 4.42 years old. Regarding the drugs consumed, the highest quantity was for illegal substances (76%), compared to the legal ones (23%). No significant difference was found between males (OR = 1.1) and females (OR = 0.74) regarding the use or abuse of multiple drugs. The elderly used both legal and illegal drugs for a long period of time, with low adherence to the treatment, and alcohol consumption among the elderly prevailed above the other psychoactive substances.

## 1. Introduction

The World Health Organization (WHO) considers the abusive use of drugs a continuous and chronic disease. According to the International Classification of Diseases (CID-10) [1], chemical substance dependence is characterized by a group of behavioral–cognitive and physiological symptoms, in which the individual continues using a drug, even when showing significant problems related to consumption [2].

The abusive use of legal and illegal substances constitutes an important public health problem [3], one that is frequently related to an individual’s life events, for instance, retirement, distancing from relatives, widowhood, social isolation, and abandonment [4].

Concerning the consumption of psychoactive substances by the population in several countries worldwide, alcohol appears to be the most used drug. Studies demonstrate that 53% of the Brazilian population has already used alcohol and 24% of these people use alcohol frequently [5]. Therefore, among the consumed substances, alcohol has demanded more attention, especially for the elderly [6].

In order to change this situation in Brazil, Psychosocial Assistance Centers for Alcohol and Drugs (CAPS-AD) were created [7], which emphasize the rehabilitation and the social reinsertion of alcohol and drug users. The psychosocial assistance is centered in community assistance, being associated with health service networks and social assistance integrated into the cultural environment [8]. The National Policy for Health Promotion (PNPS), created by Ministry of Health directive 687 2006, has as one of its priorities a reduction in mortality related to the misuse use of alcohol and other drugs. However, the PNPS does not stipulate any specific actions for the elderly, and there is no specific structure for this action [9].

In recent years, there has been a rapid and dramatic change in the Brazilian population’s age distribution. This fact was initially noticeable in developed countries, but in the last few decades, developing countries in Latin America and the Caribbean, including Brazil, have witnessed a rapid increase in the population’s aging rate [10]. Projections from the United Nations Population Fund indicate the in the year 2050 there will be more elderly people in the world than children under the age of 15 [11]. There is an expectation that in 2025, Brazil will reach the sixth world position in terms of the number of elderly people [12]. Current life expectancy for people born in Brazil is 72.74 years for men and 79.8 for women, values that may reach 77.9 for men and 84.23 for women, according to data published by the Brazilian Institute of Geography and Statistics (IBGE) [13]. As Brazil is still an underdeveloped country, it is not yet organized enough to face the new demands generated by the rise in the population’s age [10].

The Federal District, which surrounds the Brazilian Federal Capital, Brasilia, had 326,000 elderly people in the year 2016, which is about 12.8% of the total population [14].

The current study had the objective of investigating the profile of use of legal and illegal drugs and the social demographic profile of the elderly who attend one of the seven CAPS-AD units situated in the Federal District, from 2000 to 2017, thus identifying the types of drugs used, the time of use, and the probable use of associations with these drugs, as well as the adherence to the proposed treatment.

## 2. Materials and Methods

This was a quantitative, descriptive, observational, and retrospective study, with collection of secondary data from medical records, carried out in all seven CAPS-AD units in the Federal District (DF). The population mentioned in the study was formed of people who were born up to 31 December 1957 and who sought medical assistance from 2000 to 2017 in the CAPS-AD in the DF.

For the data collection from the CAPS-AD closed medical records, the following criteria of inclusion were established: (1) patients aged 60 or over; (2) who had at least had assistance for the abusive use of legal or illegal drugs; (3) with complete and understandable registered data in the medical records. A total of 650 medical records were checked, following the criteria mentioned above, from which 242 were excluded because they were incomplete, amounting to 408 medical records.

The socio-demographic variables investigated were age, marital status, education, and number of children. The types of demand regarding admission to the CAPS-AD rehabilitation services, the types of licit and illicit substances consumed, and the time of consumption and adherence to treatment were observed and presented through absolute and/or relative frequency. To compare the time of use of the different types of medication and the distribution between the sexes, the Kruskal–Wallis and Mann–Whitney tests were used, considering the significance level of 5%. For the analysis of the results, the statistical data editor Statistical Package for the Social Sciences SPSS 22^®^ (IBM, Armonk, NY, USA) was used.

The study was approved by the Ethics in Research Committee (CEP) from the Faculty of Medicine of the University of Brasilia (number of the report: 1793 889) and by the CEP from the Foundation of Teaching and Research in Health Sciences (FEPECS) in the Federal District (number of the report—1 850 877), in agreement with resolution 466/12 from the National Health Council.

## 3. Results

Most of the elderly people assisted at CAPS-AD were males (85.2%), with an average age of 64 ± 4.42 years old; married (41.4%); and with children (87.3%). The majority lived as fixed residents in houses (90.2%), and of these, 75.2% lived with their family. Nearly half of the elderly spontaneously attended the CAPS-AD (46.33%) (Table 1), and when we observed the demand for treatment between men and women, we found that women registered a higher inflow than men due to judicial orders (men = 6% and women = 8.3%).

The education of the drug users was predominantly low, where the majority had less than 5 years of study (73.53%), and 16.46% were considered illiterate (Table 1). Most of the elderly people were retired (31.37%) (*n* = 128) or unemployed (22%) (*n* = 90).

The abusive use of legal drugs (alcohol, tobacco, benzodiazepines, or psychotropics) was reported by 79% of the users. Among the illegal drugs, 21% of the users stated they used coca derivatives (cocaine, crack and merla (cocaine paste)), 14% cannabis, and 2% other drugs (solvents and shoemaker’s glue) (Figure 1).

The profile of the use of legal and illegal drugs demonstrated that alcohol was the substance most used by the elderly of both sexes who were assisted by the CAPS-AD (Table 2). The most frequent association was alcohol and tobacco, and most users started using drugs from 11 to 34 years old. Other important information about the profile of the elderly drug users can be found in Table 2.

The elderly people had, on average, 35 years of consumption of legal and or illegal drugs. Among the separately consumed drugs, alcohol was the one with the longest consumption time. The associations used more frequently involved alcohol, tobacco, cannabis, cocaine, merla, and synthetics (Table 3).

Regarding alcohol, there were no statistical differences between the time of use in decades and the groups that had more frequency of use. The time of use of psychotropics was lower when compared to the other groups of higher frequency of use (Table 4).

## 4. Discussion

The elderly who seek assistance at CAPS-AD have a lower education level and are not associated with the labor market, something that has previously been observed in other studies [15,16]. Another study in Italy showed that 64% of users had only primary education [17]. It has also been observed that once chemical dependency is installed early in life, it affects school development and the average number of years of study. The education level of the studied elderly people was low, and 74% of them did not attend elementary school. Their low education level is, indeed, a risk factor to developing chemical dependency [16].

Women seek out health services less frequently for the treatment of psychoactive substance addiction when compared to men. These results are in accordance with other research [18]. One of the reasons is that women suffer more prejudice and family pressure than men [19]. Another fact that explains a higher demand from men for rehabilitation services is because they are more affected by the problem of drug addiction than women [18]. According to Alves and Kossobudzky (2002) [19], sexual stereotypes prescribe behavioral limits for men and women, with specific demands for sexual roles, which facilitates the use of drugs by males.

Furthermore, men have a greater likelihood of access to substances, which also influences the prevalence of males in the use of substances. It is also important to emphasize that drug use may vary internationally between sexes due to the influence of culture and the accessibility policy of each location [20].

The literature has shown that elderly men usually drink more than elderly women, and women exhibit a higher probability of changing their drinking behavior as they age, a fact that may justify the low number of women in treatment [19].

CAPS-AD, in general, have become places of reference and care in the community, with the purpose of bringing users and families back into society, and they are well known by the population and health services. For this reason, it is likely that most drug users seek treatment at the CAPS-AD spontaneously, followed by a referral from health services [21].

Adults with an average age of 37.7 years old form the majority of people living in vulnerable situations on the streets, according to the census from 2015 to 2018 in Brazil, with the population over 65 years old less frequently living on the streets [20]. This is in accordance with our study, in which it was verified that most elderly people who seek CAPS-AD reported a fixed address. Housing represents an important role for quality of life, constituting a social determinant, since even healthcare is inadequate for this population [22].

The average adherence to the offered treatment at CAPS-AD is low, below 20%. Galassi et al. (2016) [23] state that patients who start the treatment only because of external influences, such as family or friends’ pressure, clinical comorbidities, and judicial orders, have difficulties in adhering to the treatment because they do not feel motivated. Moreover, difficulty in access, delay in individual care, and lack of drugs for treatment maintenance may be related to difficulty in adhesion [24]. The literature also shows that low education level is one of the characteristics of users that withdraw [17]. Studies indicate that those dependent on crack and/or other drugs adhere less to the treatment when compared to alcohol users [25]. Research developed in a chemical dependents’ rehabilitation center in the state of Paraná highlighted that alcohol dependents adhere more to the treatment, since 70% of these were discharged from psychosocial rehabilitation, against 38% of crack dependents. It was also shown that 58% of people with dependency to crack did not adhere to the treatment due to discharge caused by evasion, upon request, or for administrative discharge in cases where the patient did comply with the treatment regulations [26].

It was observed that alcohol is the most consumed drug, causing risky consumption or dependence, a fact that agrees with what was verified in studies where alcoholism has become a worldwide public health problem [1]. Social and economic costs, diseases, and health complications are observed because of the misuse of drugs. There are few studies that have directly evaluated the risk of alcohol consumption among the elderly. The literature has shown a rapid increase in the use of alcohol, especially in the elderly, in other countries such as the USA, China, and some European countries [27].

This is a particular concern, as age-related changes in metabolic, physiological, and medication profiles make the use of alcohol and other drugs more harmful among the elderly [28], increasing the likelihood of alcohol-related morbidities, potential for drug interactions, risk of intracranial hemorrhage, and basic functional limitations [29].

The international classification of diseases (ICD) defines the use of multiple drugs (ICD 10 F19) as a mental and behavioral disorder [20] that leads to the consumption of more than one drug for each individual, who often consumes them simultaneously or in sequence and usually intends to intensify, boost, or neutralize the drug’s effects [24].

In the current study, it was identified that elderly people combine more than one drug, and alcohol and tobacco are the ones most used in association. This consumption can be related to the fact that those substances are easily accessed and are freely traded in Brazil [30]. These characterization data of the studied population corroborate studies with similar percentages [16,31].

According to Oliveira and co-workers, the usage of multiple drugs can be related to long-term exposure to the use and abuse of substances, and because of the extended use, the user raises the quantity and starts to use other substances in order to have the same effects. This fact can be observed when it is noted that dependent patients make simultaneous use of other psychoactive substances, mainly tobacco, cannabis, and alcohol. It was identified that the initial age of drug consumption is between 11 and 20 years old. These data were observed after a survey from Brazilian Center of Information about Psychotropic Drugs (CEBRID) [5].

According to the third national survey on drug use by the Brazilian population, published in 2017, the median age to start general consumption was 15.7 for men and 17.1 for women, and among the under 18s, it was 13.5 [32]. Alcohol was indicated as the most consumed psychoactive substance (SPA) and was also the one consumed at the lowest age (approximately 12 years old) [33]. In a review study on alcohol consumption in 22 countries, Calvo et al. (2021) [28] verified important cross-country variation.

Alcohol dependents can be divided into two groups: “premature initiation” and “late initiation”. The premature initiation group can be characterized by alcohol dependence before old age, and the late initiation group by the development of dependence during old age. The late initiation group tends to show a dependence situation developed after crisis occasions. These individuals usually find emotional support among family and friends. In these cases, there are reports of depression, and they often try to hide the problem [34]. Corroborating our results, studies related that episodes of mental disease, such as depression, are related to the use of some drugs, especially alcohol, in older adults. Because of that, professionals’ efforts to help older adults that use abusive drugs are critical for mental health in aging populations [34]. This highlights the importance of the actions carried out by CAPS-AD.

The overwhelming majority of elderly people (85%) abandoned the treatment offered at CAPS-AD. Nevertheless, the medical records were not conclusive enough to for us to know the reasons as to why they gave up, informing us only that they attended the first assistance session and then did not attend the following assistance sessions with the reference professionals suggested by the team, such as a psychologist, a psychiatrist, and therapeutic groups. It is complex to understand the high level of evasion of this kind of treatment. Some research has studied the relationship between social consequences and engagement in treatment, but the results are mixed, with studies finding positive, negative, or no influence on treatment [35].

The Ordinance MO/MS nº 2.197/2004 [36] of the Ministry of Health established within the scope of the Unified Health System (SUS) allows, as a component of the social support network (mutual aid associations and civil society entities), the complementing of the network of services available by SUS. It means that for the person who is dependent on psychoactive substances, if they want to use any device outside the CAPS-AD performance that is a component of the SUS in a complementary way, then they are allowed, but the performance of these groups in the CAPS is not allowed; in short, the participation in Alcoholics Anonymous groups is at the discretion of the patient.

## 5. Conclusions

The present study concludes that the majority of the elderly who attend CAPS-AD are men, and that they sought the service for the abusive use of alcohol and/or tobacco, which are drugs legally marketed and with very few restrictions on their sale in Brazil. It was mostly observed that alcohol is commonly associated with other drugs, with coca, cannabis, crack, and merla being the ones with the highest incidence of association. When one observes the abusive use of illicit drugs in isolation, the use of psychotropic drugs stands out, and these were most frequently used among women who frequent CAPS-AD. Patients who use the services for the use of drugs and alcohol had used the substances for at least a decade, and the association of the use of alcohol and tobacco was found on average to be 3 decades.

This research group suggests that further studies should be carried out in order to identify the models of interventions used for the treatment, in addition to considering the effectiveness and efficiency of the treatment, together with the analysis of the structures of the call centers, given that the treatment adherence rates were less than 20%. The challenges are represented by the use of legal and illegal drugs by elderly people who started at an early age and have been users for a long time. They had experienced at least a decade of consumption, with the combination of alcohol and other psychoactive substances, in a low education level population and presenting low adherence to the treatment. The present study notes that it could have delved more specifically into the aspects of abandonment of the treatment started.

## Figures and Tables

**Figure 1 healthcare-10-00989-f001:**
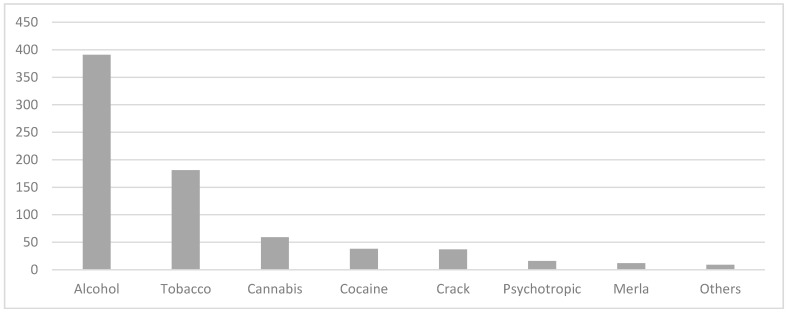
Legal and illegal drugs consumed by elderly assisted by CAPS-AD from DF between 2000 and 2017. The number of users exceeded the total amount of the sample due to the use of drugs in association.

**Table 1 healthcare-10-00989-t001:** Schooling profile, type of housing, and type of admission of elderly drug users assisted at CAPS-AD from DF between 2000 and 2017.

Variations	All Sex	Men (*n* = 348)	Woman (*n* = 60)
	(%; *n* = 408)	(%; *n* = 408)	(%; *n* = 408)
Origin			
Directed	37.00	31.13	5.88
Spontaneous	46.33	39.71	6.62
Familiar	10.29	9.31	0.98
Judicial	6.38	5.15	1.23
Type of residence			
House	75.49	65.2	10.29
Apartment	12.26	9.07	3.19
Shack	2.46	1.72	0.74
Shelter SDF *	4.66	4.17	0.49
Therapeutics community	3.19	3.19	0
Ranch	1.96	1.96	0
Education			
Illiterate	16.42	13.48	2.94
Functional illiterate	9.80	8.82	0.98
Literate	3.68	2.45	1.23
Incomplete elementary school	14.46	12.99	1.47
Complete elementary school	29.17	24.51	4.66
Incomplete high school	2.70	2.21	0.49
Complete high school	14.95	13.48	1.47
College education	8.82	7.35	1.47
Homeless *			
No	90.69	76.47	14.22
Yes	9.31	8.82	0.49

* Without fixed abode. Directed—patient referred from a health facility or referred by a healthcare professional. Spontaneous—the patient themselves sought the service for care. Familiar—a patient′s relative took him to the CAPS service. Judicial—the patient was compulsorily obliged to carry out the treatment by means of a court order.

**Table 2 healthcare-10-00989-t002:** Profile of the use of legal and illegal drugs by the elderly assisted by CAPS-AD from DF between 2000 and 2017.

Variations	All Sex	Men (*n* = 348)	Woman (*n* = 60)
	(%; *n* = 408)	(%; *n* = 408)	(%; *n* = 408)
Types of consumed drugs			
Alcohol	96.0	83.3	12.7
Tobacco	45.6	38.0	7.6
Cannabis	14.0	12.5	1.5
Coca derivatives	17.9	16.4	1.5
Psychotropics	3.4	1.2	2.2
Others	1.7	1.7	0.0
Drugs consumed separately			
Alcohol	42.4	37.7	4.6
Psychotropics	1.7	0.2	1.5
Coca derivatives	0.5	0.5	0.0
Cannabis	0.2	0.2	0.0
Main consumed associations			
Alcohol and tobacco	34	37.7	4.6
Alcohol and coca	2.0	1.7	0.3
Alcohol, tobacco, and cannabis	2.0	2.0	0.0
Alcohol, tobacco, cannabis, crack, and merla	2.0	1.7	0.2
Alcohol, cannabis, and crack	1.5	1.5	0.0
Alcohol and cannabis	1.2	1.0	0.2
Alcohol, tobacco, and psychotropics	1.0	0.5	0.5
Alcohol, tobacco, cannabis, and coca	1.2	1.2	0.0
Alcohol and crack	1.0	1.0	0.0
Alcohol, cannabis, coca, and crack	1.0	1.0	0.0
Other associations	8.3	7.3	1.0
Consumption starting age			
11 to 20 years old	35.5	31.6	3.9
21 to 30 years old	32.0	26.7	5.1
31 to 40 years old	16.4	13.5	2.9
41 to 50 years old	13.0	11.3	1.7
51 to 60 years old	2.7	2.0	0.7
Above 60 years old	0.5	0.3	0.2
Adherence to the treatment			
Yes	16.0	13.0	3.0
No	84.0	72.0	12.0

**Table 3 healthcare-10-00989-t003:** Time of consumption of several drugs and their associations among the elderly assisted by the CAPS-AD from DF between 2000 and 2017.

Type of Substance	Time in Decades
	Mean/Standard Deviation
Alcohol, tobacco, cannabis, cocaine, merla, and synthetics	4.5 ± 0.6
Alcohol, cannabis, cocaine, merla, and crack	3.5 ± 0.7
Alcohol, tobacco, and cocaine	3.33 ± 0.6
Alcohol, tobacco, and psychotropics	3.25 ± 1.5
Alcohol, tobacco, and cannabis	3.25 ± 1.16
Alcohol, tobacco, cannabis, merla, and crack	3.13 ± 1.25
Alcohol and tobacco	3 ± 1.28
Alcohol and cannabis	3 ± 1
Alcohol	2.97 ± 1.33
Alcohol, tobacco, cannabis, cocaine, and crack	2.75 ± 1.5
Alcohol and cocaine	2.75 ± 1.03
Alcohol, tobacco, cannabis, and cocaine	2.6 ± 1.34
Cocaine	2.5 ± 2.12
Alcohol, tobacco, and crack	2.5 ± 1.04
Alcohol, cannabis, cocaine, and crack	2.25 ± 0.5
Psychotropics	1.57 ± 0.98
Cannabis and crack	1.5 ± 0.7
Alcohol, cannabis, cocaine, crack, and inhalants	1.5 ± 0.7
Alcohol and crack	1.5 ± 1.3
Alcohol, cannabis, and cocaine	1.33 ± 0.6

**Table 4 healthcare-10-00989-t004:** Time of consumption in decades of substances with higher frequency of use among the elderly people assisted by the CAPS-AD in DF from 2000 to 2017.

Type of Substance	Time of Use
Alcohol (*n* = 173).	2.96 ± 1.35 ^b^
Psychotropics (*n* = 7).	1.57± 0.98 ^a,c,d,e^
Alcohol and tobacco (*n* = 139).	3 ± 1.28 ^b^
Alcohol, tobacco, and cannabis (*n* = 9).	3.25 ± 1.16 ^b^
Alcohol, cannabis, merla, crack, and tobacco time of use (*n* = 8).	3.13 ± 1.25 ^b^

^a^ = significant difference (*p* ≤ 0.05) in alcohol time of use; ^b^ = significant difference (*p* ≤ 0.05) of psychotropics time of use; ^c^ = significant difference (*p* ≤ 0.05) of alcohol and tobacco time of use; ^d^ = significant difference (*p* ≤ 0.05) of alcohol, tobacco, and cannabis time of use; ^e^ = significant difference (*p* ≤ 0.05) of alcohol, cannabis, merla, crack, and tobacco time of use.

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
