# Peer review of "Drug Use among the Elderly Assisted by the Psychosocial Assistance Center in District Federal-Brasilia"

_healthcare, 2022, doi:10.3390/healthcare10060989_

Round 1

Reviewer 1 Report

This article paints a dismal picture for those elderly with alcohol and drug problems. It reflects that the consequences of substance abuse and addiction permeate a person's life over a long period of time. We also know this has dire consequences for family members. And of course, the extremely high treatment dropout rate is very concerning. The authors should address the role of culture and its impact on rejecting or discontinuing treatment.

The sample size is good and the authors do a good job of describing the characteristics of the sample. Importantly, they tease out the various drugs that respondents used.  The relationship to low education is especially concerning.

The manuscript was prepared with much care by the authors and this is appreciated.

It would help the reader to have a more detailed description of CAPS-AD. What are the specific services that are offered to the elderly with substance abuse problems?

What is known about how the clients were referred to the program? I can't understand the top of Table 1. What do "directed," "spontaneous," and "familiar" mean in this context?

The authors note that The National Policy for Health Promotion (PNPS) does not offer specific recommendations for the treatment of the elderly (lines 55-56). This clearly should be one of the recommendations of the authors.

There is a literature on specific problems of elderly alcoholics and addicts and how to address them. The authors need to do more in the Discussion and Conclusion sections to address what could and needs to be done. Primary care doctors and nurses should be using a screening tool to assess all patients for alcohol and drug problems. Although many in the study seemed to live with family members, social isolation is overall a huge risk factor for elderly drinking and drug use. Can the authors offer some suggestions for engaging the family in treatment? How about referrals to 12-Step groups such as Alcoholics Anonymous?

Overall, the article does a good job of bringing serious substance abuse problems to light. However, the authors can do more to offer suggestions for helpful interventions.

Author Response

We were very pleased with the feedback from reviewers. They made pertinent comments and several plausible changes. Each comment was answered in a letter and its changes were made in the manuscript.

Reviewer 2 Report

Thank you for the opportunity to review this manuscript.  It is well done overall.  This reader have minor recommendations regarding choice of words that may come across as bias or stigmatizing such as "abusive use of drugs".   The reader recommends replacing with the word "misuse of drugs".  The sentence on line 50-51 "which emphasize the rehabilitation and the social reinsertion of alcohol and drug users" is not clear to the reader. 

Materials and Methods.  

The inclusion criteria of patients age 60 and older is incongruent with the birth date parameter of Dec 31, 1956.  This date will make the patients 65 years old.   Is it possible to include a graphic representation of the inclusion criteria? No need to but a visual may be helpful to the reader.

Discussion.

Line 157.  "..affected by the problem of drug addiction.."  It may be necessary to add a statement to expand on how men are affected more by drug addiction.  Line 169 word "known" is repeated.  Line 191 in an effort to reduce biased language, this reader recommends replacing "..58% of the crack dependents.." with ..people with dependency to crack.. ; Line 197.  replace "abusive behavior" with ..use pattern..

Conclusions.

The reader recommends adding a statements of the limitations of the study.

Author Response

(The authors gave the same response as above.)

Reviewer 3 Report

In this study, the authors investigated the profile of use of legal and illegal drugs by the elderly who are assisted by the network of psychosocial assistance in the Federal District of Brasil. The authors stated that the elderly used both legal and illegal drugs for a long time, with low adherence to the treatment, and alcohol consumption among the elderly prevails above the other psychoactive substances. In their conclusions, the authors speculated that "further studies should be carried out in order to identify the models of interventions used for the treatment given that the treatment adherence rates are less than 20%. The challenges, according to the authors, are represented by the use of legal and illegal drugs by elderly people, who started at an early age, and have been users for a long time".

The paper is interesting however, is hard to find for the reader novelties in their investigation.

I do suggest also comparing their data with findings originating in other countries do not limit their conclusions at a national/regional level.

Author Response

(The authors gave the same response as above.)

Round 2

Reviewer 3 Report

The paper is easy to follow